# Hypothetical Proteins of *Mycoplasma synoviae* Reannotation and Expression Changes Identified via RNA-Sequencing

**DOI:** 10.3390/microorganisms11112716

**Published:** 2023-11-06

**Authors:** Duoduo Si, Jialin Sun, Lei Guo, Fei Yang, Xingmiao Tian, Shenghu He, Jidong Li

**Affiliations:** College of Animal Science and Technology, Clinical Veterinary Laboratory, Ningxia University, Yinchuan 750021, China; s956173936@163.com (D.S.); 18300795598@163.com (J.S.); guoleinxyc@163.com (L.G.); yangfeiweiwuxian@126.com (F.Y.); 13935097276@163.com (X.T.)

**Keywords:** *Mycoplasma synoviae*, hypothetical proteins, reannotation, virulence factors, expression identification

## Abstract

*Mycoplasma synoviae* infection rates in chickens are increasing worldwide. Genomic studies have considerably improved our understanding of *M. synoviae* biology and virulence. However, approximately 20% of the predicted proteins have unknown functions. In particular, the *M. synoviae* ATCC 25204 genome has 663 encoding DNA sequences, among which 155 are considered encoding hypothetical proteins (HPs). Several of these genes may encode unknown virulence factors. This study aims to reannotate all 155 proteins in *M. synoviae* ATCC 25204 to predict new potential virulence factors using currently available databases and bioinformatics tools. Finally, 125 proteins were reannotated, including enzymes (39%), lipoproteins (10%), DNA-binding proteins (6%), phase-variable hemagglutinin (19%), and other protein types (26%). Among 155 proteins, 28 proteins associated with virulence were detected, five of which were reannotated. Furthermore, HP expression was compared before and after the *M. synoviae* infection of cells to identify potential virulence-related proteins. The expression of 14 HP genes was upregulated, including that of five virulence-related genes. Our study improved the functional annotation of *M. synoviae* ATCC 25204 from 76% to 95% and enabled the discovery of potential virulence factors in the genome. Moreover, 14 proteins that may be involved in *M. synoviae* infection were identified, providing candidate proteins and facilitating the exploration of the infection mechanism of *M. synoviae*.

## 1. Introduction

*Mycoplasma synoviae* is a bacterium within the Mollicutes class that lacks a cell wall and causes respiratory damage, infectious synovitis, and arthritis in chickens. *M. synoviae* is a devastating pathogen of chickens that costs the poultry industry billions of dollars each year [1].

*Mycoplasma synoviae* infection in chickens is susceptible to secondary infection by other pathogenic microorganisms, leading to the manifestation of atypical clinical symptoms during the initial stages. At present, the understanding of the pathogenesis of *M. synoviae* in chickens remains incomplete, encompassing both host and bacterial factors as key contributors to its pathogenicity. The primary determinants of its host are the innate and adaptive immune responses, such as the interaction between *M. synoviae* and chicken synovial sheath cells (SSCs), which contribute to the inflammatory response through the upregulation of cytokines and the attraction of macrophages [2]. Secondary host factors are somatic changes that affect the execution of normal cell functions, such as CCH deformation, which increased cytokine gene expression, and extensive metabolic and sensitivity changes in cells when exposed to *M. synoviae* [3]. Bacterial virulence factors play a crucial role in various processes, such as the adhesion to host cells, maintenance of homeostasis, invasion of host cells, and regulation of the immune response. These factors significantly impact the colonization, immunogenicity, and transmissibility of pathogens within the host [4]. The known virulence factors of *M. synoviae* in chickens are adhesins, proteases, and membrane transporters, which regulate several biological processes, such as cell adhesion, overall metabolism, and host–pathogen interactions [4,5,6]. Highly variable virulence factors of *M. synoviae* contribute to immune escape [7]. In addition to these virulence factors, uncharacterized proteins have the potential to perform virulence-related functions.

The *M. synoviae* genome is 846,495 bp in length, with a G + C content of 28.3%, on a mol basis, and 34.2% estimated by the buoyant density [8]. The genome encodes 673 proteins, 10 of which are repeated and 155 annotated as hypothetical proteins (HPs). Many proteins are involved in *M. synoviae* infection. Dihydrolipoamide dehydrogenase, NADH oxidase, and the pyruvate dehydrogenase complex (PDC) E1 alpha and beta subunits of *M. synoviae* are all immunogenic, may bind the fibronectin/plasminogen protein, and are involved in the host adhesin process [9,10]. Oligopeptide (Opp) permease, recognized for its participation in humoral immune responses, can be researched as a potential candidate antigen [11]. NADH oxidase participates in *M. synoviae* adhesion to host cells can be studied as a diagnostic antigen and a potential protective vaccine candidate [6]. However, the functions of the HPs are unknown. We hypothesize that the reannotation of unannotated coding sequences (CDSs) in the *M. synoviae* genome may lead to the discovery of novel structures and functions and provide new ideas and insights into protein-interaction network pathways. Furthermore, such novel structures and functions may become new targets for drug design.

Therefore, this study analyzes the 155 unknown HPs for function predictions using bioinformatics web tools, including BLAST, INTERPROSCAN, PFAM, and COGs. The accuracy of these tools is verified by the receiver operating characteristic (ROC) curve analysis. The expression of HPs is identified by RNA-Seq. Among those HPS, 28 are predicted as virulence proteins, including 19 virulence proteins, 125 are reannotated, and 9 virulence proteins are not reannotated. Fourteen HP genes are significantly upregulated, nine of which are reannotated, and five are determined as putative virulence genes. Overall, an improved understanding of the biology and pathogenicity of *M. synoviae* is achieved as a result of these newly discovered virulence factors. The upregulated genes may help in developing strategies to control *M. synoviae* infection.

## 2. Materials and Methods

### 2.1. HPs’ Amino Acid Sequence Retrieval

The sequences of *M. synoviae* HPs were downloaded from the NCBI database (https://www.ncbi.nlm.nih.gov/nuccore/NZ_CP011096.1, accessed on 13 July 2021) and 155 identified HPs were analyzed.

### 2.2. HPs’ Physicochemical Properties and Subcellular Localization Predictions

Physicochemical properties and subcellular localization predictions were achieved using a variety of tools summarized in Table 1.

### 2.3. Function Prediction of HPs

The tools used for predicting the HPs’ functions are listed in Table 2.

### 2.4. Accuracy Assessment of Tools

Fifty known proteins randomly selected from the NCBI protein library were annotated using each tool (Appendix A). The results of applying each tool to each protein were assigned, as previously described [33], and the accuracy of each tool was evaluated using the ROC (http://www.rad.jhmi.edu/jeng/javarad/roc/JROCFITi.html, accessed on 2 March 2023).

### 2.5. Prediction of HPs’ Association to Virulence

The virulence association of HPs was characterized using the Virulent Pred server. The virulence factor database (VFDB) [34] was analyzed to determine the presence of any potential orthologs for the predicted virulence factors. BTX Pred [35] and DBETH [36] were used to analyze the cytotoxic potential of HPs, and CARD software (Version 3.2.5) [37] was used to analyze their association to antibiotic resistance.

### 2.6. Prediction of HPs’ Antigenicity

An antigenicity analysis was performed based on the location of HPs predicted to reside in cytoplasmic membranes and/or extracellular milieu. An antigenicity prediction tool was used to assess the epitopes of the HPs (http://imed.med.ucm.es/Tools/antigenic.pl, Accessed on 2 November 2022). The epitopes were retrieved and their lengths were measured to determine the epitope coverage. Antigenic HPs were subsequently identified based on the corresponding epitope predictions.

### 2.7. Changes in HP Expression Values upon Exposure to Chicken Cells

#### 2.7.1. The Source of the RNA-Seq Data

The data used for the HP expression analysis were derived from the prokaryotic transcription data obtained from our laboratory. MS-Host1, MS-Host2, and MS-Host3 samples were *M. synoviae* exposed to a mixture of chicken macrophages (HD-11) and chicken embryo fibroblasts (DF-1); MS-1, MS-2, and MS-3 were *M. synoviae* cultured in vitro (Appendix A).

#### 2.7.2. RNA-Seq Analyses of HPs

An RNA-Seq approach was used to explore the changes in *M. synoviae* HP expression after the exposure of DF-1 and HD-11 cells to *M. synoviae*. Differentially expressed HP genes (DEHPGs) were analyzed using Omicshare (https://www.omicshare.com/tools/Home/Soft/diffanalysis, accessed on 9 March 2023). Gene Ontology (GO) enrichment analyses were also conducted using Omicshare (https://www.omicshare.com/tools/Home/Soft/gogseasenior, accessed on 10 March 2023), with *p* < 0.05 corresponding to a significant GO term enrichment. The KEGG database (http://www.kegg.jp/show_organism?menu_type=pathway_map&org=mbi, accessed on 10 March 2023) was used to analyze the pathways for which DEHPGs were enriched.

## 3. Results

### 3.1. Hypothetical Proteins Encoded by the M. synoviae ATCC-25204 Genome

The *M. synoviae* ATCC 25204 genome consists of 720 CDSs and 673 genes, among which 155 encode HPs. However, only 8.39% of all HPs (13 out of 155) have been detected in membrane-associated lipoprotein proteomic studies [38]. 

### 3.2. Physicochemical Properties and Subcellular Localization of M. synoviae ATCC-25204 HPs

The physicochemical properties of all the HPs were predicted (Appendix A). These HPs ranged from 45–1575 amino acid residues in size and had molecular weights ranging from 5.2–184.2 kDa. The isoelectric points of the HPs ranged from 4.28–11.62 and 66% of these proteins had an isoelectric point above pH 7.0. The grand average hydropathicity index of HPs ranged from −1.077 to 1.067, with 85% of them having a negative index. The extinction coefficients of the HPs ranged from 1490–426,540. Of the HPs, 86% were predicted to be stable according to the instability index. Furthermore, the predicted subcellular localization of 155 HPs (Appendix A) resulted in 67 assigned to the cytoplasm (~43.23%) and 37 assigned to the cytoplasmic membrane (~23.87%), and 52 were putative extracellular proteins (~33.5%) (Figure 1). Nineteen and 33 extracellular HPs were considered to be secreted by classical and non-classical pathways, respectively.

### 3.3. Putative Function of M. synoviae ATCC 25204 HPs

Hypothetical proteins were functionally annotated using the GO categorization, identifying homologs, and predicting functional domains and partners. A total of nine and five proteins were assigned to the subcategories ‘activity’ and ‘binding’, respectively, among the HPs categorized under molecular function (MF). DNA binding was the most represented subcategory in the MF category (3 out of 13 proteins), followed by ‘N-methyltransferase activity’ (2 out of 13 proteins). The biological process (BP) subcategories were assigned to three HPs, and two HPs were related to DNA methylation (Appendix A). Eleven HPs had hits for functional homologs predicted by the three used tools (Appendix A). After the functional homologs were identified, functional domains and partners were predicted for HPs (Appendix A). Finally, putative functional partners were predicted for the 155 HPs (Appendix A). Altogether, a function was assigned to 125 out of 155 HPs (80.65%, Appendix A). The most represented putative functional groups were enzymes, lipoproteins, DNA-binding proteins, and phase-variable hemagglutinins (Figure 2). Forty-nine proteins were predicted to have an enzymatic function; most of them were thought to be functional enzymes, accounting for 20% of the PPFs. These enzymes belonged to the five major enzyme groups of transferases, synthetase, hydrolases, isomerases, and oxidoreductases. Among them, transferase accounted for the highest proportion, followed by synthetase. We also found that two enzymes, recombinase and permease, were not listed among the seven enzymes (Table 3). Moreover, 24 proteins were presumed to be hemagglutinin (Table 4). Thirteen proteins were presumed to be lipoproteins (Table 5). Seven proteins were presumed to have DNA-binding proteins (Table 6). Thirty-two proteins were listed in other functional categories, three proteins were predicted to be aromatic cluster surface proteins, two were thought to be lipocalins, and an elongation factor EF-Ts was annotated this time (Table 7). Whether the elongation factor EF-Ts was virulent was also revealed by a subsequent analysis. These HPs were considered proteins with putative functions (PPFs). Our results improve the annotation status of the *M. synoviae* ATCC 25204 genome from 76% to 95%.

### 3.4. Accuracy Assessment of Functional Prediction Tools

The mean accuracy value of the tools used for this study was 0.95 (Appendix A), indicating that the prediction results obtained for *M. synoviae* ATCC-25204 HPs were reliable.

### 3.5. Prediction of HP Genes’ Association to the Virulence of M. synoviae

Based on VirulentPred, a total of 149 proteins among the 155 (96.13%) comprising the entire set of *M. synoviae* ATCC 25204 HPs (Appendix A) were considered as putative virulence factors. In silico predictions of putative associations with virulence identified at least 28 novel potential virulence-related proteins, combined with the VFDB. These 28 putative virulence factors included 11 enzymes (VY93_RS00915, VY93_RS00970, VY93_RS02560, VY93_RS01570, VY93_RS00275, VY93_RS03120, VY93_RS01815, VY93_RS00470, VY93_RS00400, VY93_RS00850, and VY93_RS02300), three lipoproteins (VY93_RS01720, VY93_RS00960, and VY93_RS01825), one phase-variable hemagglutinin (VY93_RS01280), one replication initiation and membrane attachment protein (VY93_RS02865), one aromatic cluster surface protein (VY93_RS00705), one SMC-like protein (VY93_RS02250), one PotD/PotF family extracellular solute-binding protein (VY93_RS02790), and ten proteins with unknown functions (VY93_RS01950, VY93_RS02530, VY93_RS03220, VY93_RS01855, VY93_RS02530, VY93_RS02755, VY93_RS00345, VY93_RS01755, VY93_RS00115, and VY93_RS02740). Additionally, no potential virulence factor was assessed as a cytotoxic protein. Overall, the high number of predicted putative virulence factors may have reflected their relevance to *M. synoviae* homeostasis.

### 3.6. Prediction of Antigenicity 

Following the antigenicity analyses of the HPs, a set of 89 HPs predicted to be localized in the cytoplasmic membrane and/or produced as extracellular proteins was used to perform an epitope prediction. All the analyzed HPs presented at least one predicted epitope, with the epitope numbers ranging from 1–53 (Appendix A). Moreover, these predicted epitopes covered 20.33–84.05% of the protein length. The potential virulence factors presented at least three predicted epitopes, with the epitope numbers ranging from 39–552 and epitope cover from 25.19–56.10%.

### 3.7. Identification of Differentially Expressed HP Genes 

Changes in the HP transcriptome were evident following the infection of chicken cells (Appendix A). Furthermore, the transcriptome analysis revealed that 45 out of 83 genes were upregulated (Figure 3).

### 3.8. GO and Pathway Enrichment Analyses

Initially, 51 of the 155 HP genes (32.90%) were categorized according to GO terms into ‘cellular component’. Furthermore, among all the HP genes, the expressions of 10 were upregulated, and 139 were categorized into the ‘biological process’ group (Appendix A).

Most DEGs associated with these GO categories were downregulated, and a significant enrichment of the nucleotide-binding pathway was observed. Moreover, the top 25 GO categories that were enriched with upregulated DEGs (upregulated DEGs/total DEGs enriched in a GO category) included developmental processes, sulfur amino acid transmembrane transporter activity, and tRNA binding. Most latent virulence factors and pathogenic effector genes were enriched in the cellular component category (Figure 4). Proteins VY93_RS02865, VY93_RS01280, VY93_RS00400, VY93_RS01815, and VY93_RS00275 among the 28 putative virulence-factor genes were enriched in 19, 18, 12, 7, and 3 GO categories, respectively.

The KEGG pathway analysis of the identified the HPs revealed their enrichment in 99 pathways (Appendix A). The ribosome pathway was the most enriched (51 genes), followed by the ABC transporters (32 genes) and aminoacyl-tRNA biosynthesis (24 genes) pathways. Other highly enriched pathways included glycolysis/gluconeogenesis, quorum-sensing, oxidative phosphorylation, homologous recombination, photosynthesis, pyrimidine metabolism, DNA replication, mismatch repair, pentose phosphate pathway, urine metabolism, and protein export pathways, which contained 18, 17, 14, 14, 13, 12, 12, 11, 10, 10, and 10 genes, respectively. A total of 28 putative virulence-factor genes were subjected to the pathway enrichment analysis, as a result of which VY93_RS00275 and VY93_RS00400 were associated with mismatch repair, DNA replication, and homologous recombination pathways (Figure 5).

## 4. Discussion

In this study, we used previously predicted in silico approaches to reannotate *M. synoviae* ATCC-25204 HPs and to identify potential virulence-related proteins [39]. 

The physicochemical properties of these HPs indicated that they had high heterogenicity results, as they varied over wide ranges concerning the factors of length, molecular weight, theoretical isoelectric point, grand average of hydropathicity, and extinction coefficient. These HPs seemingly encoded different functional products, given their divergent intrinsic sequence properties. These characteristics are likely to have contributed to their distinct functions [40]. Most of these proteins were predicted to be stable in vitro, a finding that indicated their potential for heterologous protein expression, which enabled functional and/or immunological studies in vitro [41].

The subcellular localization predictions indicated that *M. synoviae* HPs would be found in the cytoplasm, cytoplasmic membrane, or extracellular fractions. Some HPs predicted to be both cytoplasmic and extracellular proteins were found in the surface fraction in our previous proteomic study [38]. Therefore, some HPs were also located in multiple compartments and/or served moonlight functions [42]. Furthermore, approximately 63.46% of the extracellular HPs were secreted by non-classical pathways, suggesting that they may have been secreted via membrane vesicles [43]. Overall, >57% of the HPs were predicted as surface or extracellular fractions, indicating that they may have been relevant to *M. synoviae* pathogenesis and were potentially involved in host–pathogen processes.

The GO classification, functional homolog search, and the predictions of functional domains and partners were combined based on the selected functional domains to assign *M. synoviae* HPs’ putative functions. Thus, different prediction strategies with overlapping assigned functions to HPs supported their functional annotations. Moreover, each strategy identified the functions unassigned by other prediction methods. Therefore, the *M. synoviae* ATCC 25204 genomic annotation was improved by these analyses, and many diverse functional groups were found to contain enzymes (38.58% of PPFs), lipoproteins (10.24% of PPFs), DNA-binding proteins (5.51% of PPFs), and phase-variable hemagglutinin (19.69% of PPFs). The assignment of the putative functional roles for HPs provides new insights into *M. synoviae* biology, including the homeostasis and pathogenesis of bacteria. 

We speculated that these proteins had an association with several relevant functions in *M. synoviae*, varying from various growth stress responses and virulence mechanisms to bacterial survival when exposed to the host [44]. Additionally, several proteins were identified as putative proteases. Thus, for example, these enzymes played important roles in the adhesion of mycoplasmas to the host and its immunomodulatory effects [45]. Phenylalanyl-tRNA synthetase and NADH-ubiquinone oxidoreductase were related to bacterial growth or responses to various growth conditions [46,47]. Cell division and virulence were facilitated by intracellular septation proteins [48]. All these proteins played an important role in cell envelope biogenesis, maintaining the integrity of the cell envelope and ensuring membrane homeostasis [49,50,51,52]. In addition to the putative functions associated with bacterial growth and homeostasis, the putative functions related to *M. synoviae* virulence were also predicted. A case in point, lipoproteins are known to play an important role in infection and do not cause clinical symptoms. Additionally, the immune system uses them to promote inflammation and leukocyte recruitment to infected tissues [53]. Hemagglutinins, which mediate the adhesion of mycoplasma to cells, are encoded by the *M. synoviae vlhA* gene, which belongs to a reservoir of pseudogene alleles. Variants of vlhA are expressed from the same unique vlhA promoter by recruiting pseudogene sequences via site-specific recombination events to generate antigenic variability [54]. Each expresses a different allele of vlhA as an antigen-diversifying mechanism to evade the adaptive immune system of the host [55]. Among them, 25 hemagglutinins were identified in this study, 1 of which was detected in an upregulated state and was believed to participate in the adhesion to and invasion of host cells by *M. synoviae*. 

Moreover, these putative proteins had potential antigenic properties mainly associated with the regulation of host immune responses as potential mechanisms of antigenic variation [56]. This study found that there were many PPFs generally associated with *M. synoviae* virulence, although *M. synoviae* survival depended on certain putative functions. 

In the entire HP transcriptome, VY93_RS02530, VY93_RS03220, VY93_RS00345, and VY93_RS02935 were differentially expressed when infecting host cells and were predicted as bacterial virulence factors, although they were not annotated. Putative aromatic cluster surface proteins (VY93_RS00715 and VY93_RS00710), which played a role in protein–protein interactions [57], were also significantly upregulated. Furthermore, RNA polymerase, an essential enzyme for bacterial viability, was also found to be upregulated in this study. Finally, intracellular septation proteins were also upregulated, indicating that these proteins may have participated in *M. synoviae* proliferation [58]. Thus, all upregulated proteins may be involved in the invasion and pathogenicity of *M. synoviae*. The specific functions of these proteins during *M. synoviae* infection and their effect on the host require verification through further experimentation.

## 5. Conclusions

In this study, the *M. synoviae* ATCC-25204 genome was reannotated and an increase in protein-coding CDSs was observed. A total of 125 proteins were functionally annotated and were predicted to be virulence factors. Among the 155 HP genes, 14 were upregulated upon exposure to host cells, including several genes that promoted *M. synoviae* division and growth. Furthermore, five of the 28 potential virulence factors were also upregulated, and these proteins were predicted to play a crucial role in *M. synoviae* infection. However, the elucidation of the actual roles of the novel *M. synoviae* potential virulence factors requires further experimental work. Overall, our findings contribute to developing new strategies for treating, preventing, and controlling infections caused by *M. synoviae*.

## Figures and Tables

**Figure 1 microorganisms-11-02716-f001:**
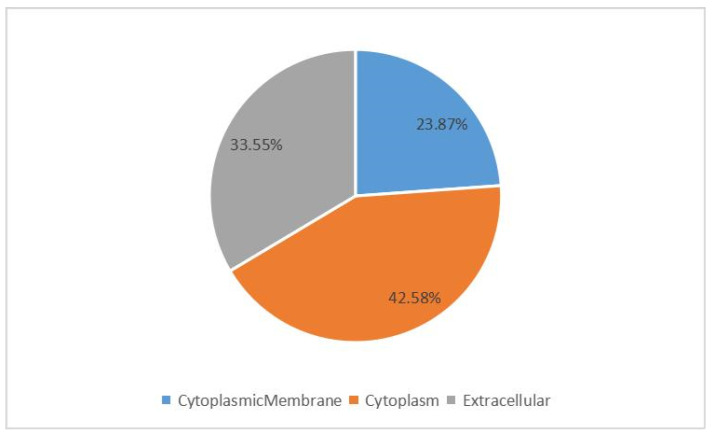
Prediction of subcellular localization of *M. synoviae* ATCC 25204 hypothetical proteins. Percentages of proteins assigned to a given subcellular localization (cytoplasm, cytoplasmic membrane, or extracellular) are expressed relative to the total number of analyzed proteins.

**Figure 2 microorganisms-11-02716-f002:**
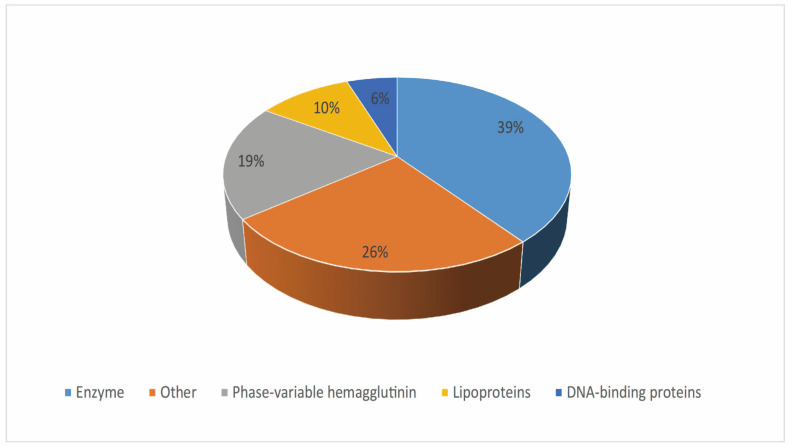
Distribution of the main functional groups assigned to *M. synoviae* ATCC-25204 hypothetical proteins. Percentages of PPFs assigned to ‘enzymes’, ‘phase-variable hemagglutinins’, ‘lipoproteins’, and ‘DNA-binding proteins’; functional groups are indicated. Some proteins were assigned to more than one functional group.

**Figure 3 microorganisms-11-02716-f003:**
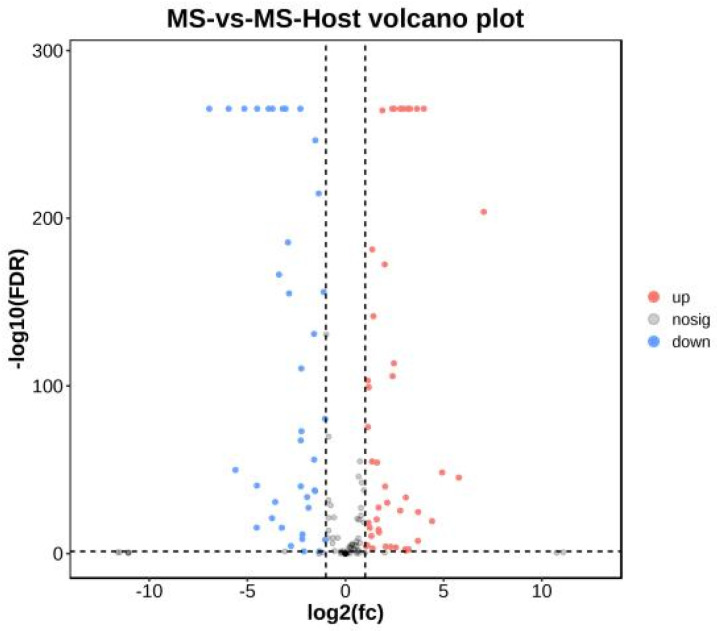
Volcano plot of differentially expressed hypothetical protein genes. Cutoff for log2(fc) is 1; the numbers of upregulated and downregulated genes were 45 and 38, respectively; The red dots represent significantly upregulated genes, blue dots represent significantly downregulated genes, and grey dots represent non-significant genes.

**Figure 4 microorganisms-11-02716-f004:**
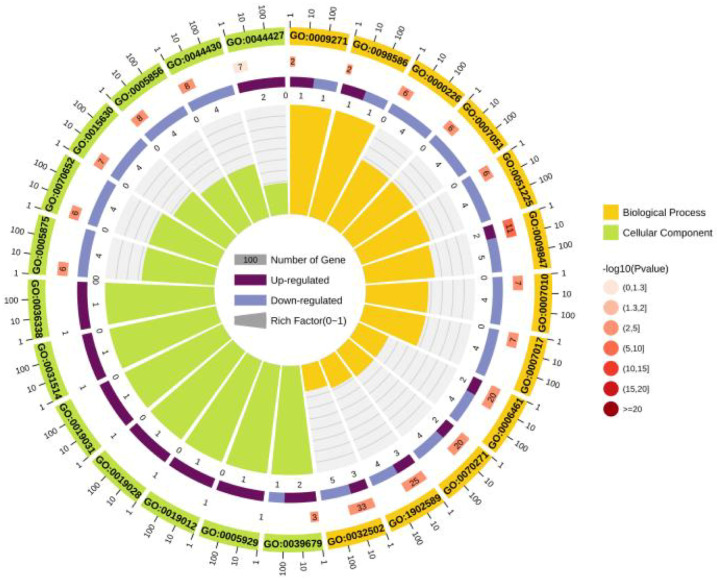
GO analysis of differentially expressed hypothetical protein genes.

**Figure 5 microorganisms-11-02716-f005:**
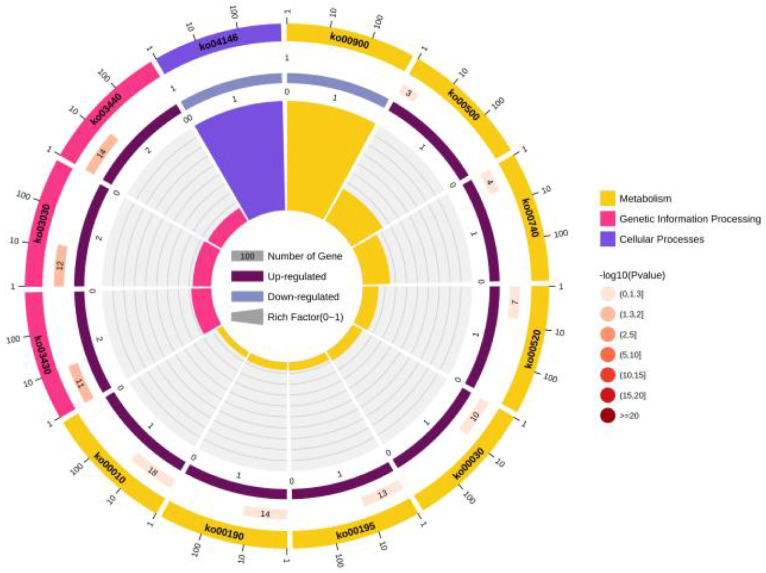
KEGG pathway analysis of differentially expressed hypothetical protein genes.

**Table 1 microorganisms-11-02716-t001:** Bioinformatics tools used for the determination of physicochemical properties and subcellular localization predictions of *M. synoviae* hypothetical proteins.

Name	URL	Reference
ExPASy-ProtParam	https://web.expasy.org/protparam/, accessed on 3 September 2022	[12]
PSORTb	https://www.psort.org/psortb/, accessed on 5 September 2022	[13]
PSLPred	http://crdd.osdd.net/raghava/pslpred/, accessed on 6 September 2022	[14]
LOCTree3	https://rostlab.org/services/loctree3/, accessed on 7 September 2022	[15]
HMMTOP	http://www.enzim.hu/hmmtop/, accessed on 8 September 2022	[16]
TMHMM	http://www.cbs.dtu.dk/services/TMHMM/, accessed on 9 September 2022	[17]
Phobius	https://phobius.sbc.su.se/, accessed on 13 September 2022	[18]
SignalP	http://www.cbs.dtu.dk/services/SignalP-3.0/, accessed on 15 September 2022	[19]
SecretomeP	http://www.cbs.dtu.dk/services/SecretomeP/, accessed on 15 September 2022	[20]

**Table 2 microorganisms-11-02716-t002:** Bioinformatics tools used for the function prediction of *Mycoplasma synoviae* hypothetical proteins.

Name	URL	Reference
BLAST2GO	https://www.blast2go.com/, accessed on 25 September 2022	[21]
HMMER	https://www.ebi.ac.uk/Tools/hmmer/, accessed on 29 September 2022	[22]
FASTA	https://fasta.bioch.virginia.edu/fasta_www2/fasta_www.cgi, accessed on 29 September 2022	[23]
Pfam	https://pfam.xfam.org/, accessed on 5 October 2022	[24]
SUPERFAMILY	http://www.supfam.org/SUPERFAMILY/, accessed on 6 October 2022	[25]
CATH	https://www.cathdb.info/, accessed on 11 October 2022	[26]
CDART	https://www.ncbi.nlm.nih.gov/Structure/lexington/lexington.cgi, accessed on 15 October 2022	[27]
SMART	https://smart.embl-heidelberg.de/, accessed on 17 October 2022	[28]
SBASE	http://pongor.itk.ppke.hu/protein/sbase. html#/sbase_blast, accessed on 19 October 2022	[29]
InterPro	https://www.ebi.ac.uk/interpro/, accessed on 11 October 2022	[30]
STRING	https://string-db.org/, accessed on 23 October 2022	[31]
STITCH	http://stitch.embl.de/, accessed on 29 October 2022	[32]

**Table 3 microorganisms-11-02716-t003:** *M. synoviae* ATCC 25204 PPFs predicted as putative enzymes.

NCBI Gene ID	Putative Function
	Transferases
VY93_RS00970	DNA polymerase; luciferase; phosphatidate Cytidylyltransferase
VY93_RS02560	Protein kinase
VY93_RS01570	Protein kinase
VY93_RS03110	Riboflavin kinase
VY93_RS00830	Sensor kinase
VY93_RS00850	Phosphotransferase; hemagglutinin
VY93_RS01710	Glutathione S-transferase
VY93_RS02970	Mycoplasma MFS transporter; histidine kinase A
VY93_RS00115	Transferase (glycosyl, DHHC palmitoyl); histidine kinase
VY93_RS01810	Membrane-bound O-acyl transferase
VY93_RS02170	tRNA/rRNA methyltransferase
VY93_RS02255	Protein kinase
VY93_RS03860	RNA-binding S4 domain-containing protein; protein kinase
VY93_RS00470	Protein kinase
VY93_RS01555	ATP-dependent protease; BTB protein DNA polymerase
VY93_RS00275	DNA polymerase III
VY93_RS01930	ParB-like nuclease; DNA-directed RNA polymerase
VY93_RS00350	RNA polymerase
VY93_RS00400	DNA polymerase III
VY93_RS02300	Methyltransferase type 12
VY93_RS03705	DNA methylase; S-adenosyl-L-methionine-dependent methyltransferase; typeIII restriction-modification system StyLTI enzyme
	Synthetase
VY93_RS03395	Phenylalanyl-tRNA synthetase
VY93_RS03120	tRNA pseudouridine synthase B
VY93_RS01715	Copper-transporting ATPase
VY93_RS00840	ATP synthase
VY93_RS01815	Phox homology (PX) domain protein; cysteinyl-tRNA synthetase
VY93_RS04225	DNA ligase
VY93_RS02000	Mur ligase
VY93_RS03720	ATPase
VY93_RS04185	ATPase
VY93_RS00770	IVS-encoded protein-like superfamily; AAA ATPase
	Hydrolases
VY93_RS01095	Primase-polymerase; phosphohydrolase
VY93_RS02860	P-loop containing nucleoside triphosphate hydrolases
VY93_RS02175	Phosphomevalonate kinase; P-loop containing nucleoside triphosphate hydrolase
VY93_RS03870	Clostridium epsilon toxin ETX; bacillus mosquitocidal toxin MTX2; deoxyribonuclease I
VY93_RS03200	Alkaline phosphatase
VY93_RS02315	Peptidase M32
VY93_RS02730	Peptidase M13
VY93_RS00915	Peptidase C39
VY93_RS04205	Ribonuclease H
VY93_RS02305	Bifunctional 2′,3′-cyclic-nucleotide 2′-phosphodiesterase/3′-nucleotidase; bacterial extracellular solute-binding protein; fibronectin-binding protein
VY93_RS03875	DNA methylase
	Isomerases
VY93_RS02425	Isomerase (sugar; phosphoglucose)
VY93_RS01705	Topoisomerase-primase
VY93_RS00205	Galactose mutarotase
	Oxidoreductases
VY93_RS00910	NADH-ubiquinone oxidoreductase
VY93_RS02220	Acyl-CoA oxidase; DNA methylase
	Others
VY93_RS03020	Recombinase Flp protein
VY93_RS02085	Permease (ABC-type glycerol-3-phosphate transport system; carbohydrate ABC transporter)

**Table 4 microorganisms-11-02716-t004:** *M. synoviae* ATCC 25204 PPFs predicted as hemagglutinin.

NCBI Gene ID	Putative Function
VY93_RS00850	Phosphotransferase; hemagglutinin
VY913_RS01400	Phase-variable hemagglutinin
VY93_RS03820	Phase-variable hemagglutinin
VY93_RS01245	Phase-variable hemagglutinin
VY93_RS01250	Phase-variable hemagglutinin
VY93_RS01255	Phase-variable hemagglutinin
VY93_RS01260	Phase-variable hemagglutinin
VY93_RS01280	Phase-variable hemagglutinin
VY93_RS01285	Phase-variable hemagglutinin
VY93_RS04265	Phase-variable hemagglutinin
VY93_RS04120	Phase-variable hemagglutinin
VY93_RS01315	Phase-variable hemagglutinin
VY93_RS01350	Phase-variable hemagglutinin
VY93_RS01420	Phase-variable hemagglutinin
VY93_RS01425	Phase-variable hemagglutinin
VY93_RS01450	Phase-variable hemagglutinin
VY93_RS01455	Phase-variable hemagglutinin
VY93_RS01380	Phase-variable hemagglutinin
VY93_RS04325	Phase-variable hemagglutinin
VY93_RS01460	Phase-variable hemagglutinin
VY93_RS01270	Phase-variable hemagglutinin
VY93_RS01330	Phase-variable hemagglutinin
VY93_RS01410	Phase-variable hemagglutinin
VY93_RS00340	Phase-variable hemagglutinin
VY93_RS03730	Phase-variable hemagglutinin

**Table 5 microorganisms-11-02716-t005:** *M. synoviae* ATCC 25204 PPFs predicted as lipoprotein.

NCBI Gene ID	Putative Function
VY93_RS00960	P60-like lipoprotein
VY93_RS04200	Membrane lipoprotein
VY93_RS01720	Membrane lipoprotein
VY93_RS00485	P37-like ABC transporter substrate-binding lipoprotein
VY93_RS03535	Lipoprotein
VY93_RS01895	Lipoprotein
VY93_RS01900	Lipoprotein
VY93_RS02090	Lipoprotein
VY93_RS00965	Membrane protein P80
VY93_RS01825	Murein lipoproteins
VY93_RS04030	Mycoplasma lipoprotein (MG045)
VY93_RS00405	Mycoplasma lipoprotein; fimbrial protein
VY93_RS00700	Membrane protein

**Table 6 microorganisms-11-02716-t006:** *M. synoviae* ATCC 25204 PPFs predicted as putative DNA-binding proteins.

NCBI Gene ID	Putative Function
VY93_RS00780	Putative helix-turn-helix protein (YlxM/p13-like)
VY93_RS00990	Ribbon-helix-helix protein
VY93_RS01500	Transcription factors
VY93_RS00780	Putative helix-turn-helix protein (YlxM/p13-like)
VY93_RS01550	Nucleic acid binding
VY93_RS03080	Transcriptional regulatory protein
VY93_RS00845	Guanine nucleotide exchange factor (GEF) domain of SopE; Pleckstrin homology-related domain protein

**Table 7 microorganisms-11-02716-t007:** *M. synoviae* ATCC 25204 PPFs predicted as putative other functions.

NCBI Gene ID	Putative Function
VY93_RS01990	Nematode chemoreceptor
VY93_RS04250	Elongation factor EF-Ts
VY93_RS00240	Seadorna_VP6
VY93_RS03025	Pore-forming protein, afimbrial adhesin AFA-I
VY93_RS02525	Cadherins; anticodon_Ia_like protein
VY93_RS02420	Smr protein
VY93_RS01110	Super-infection exclusion protein B;
VY93_RS00790	Transmembrane protein 51
VY93_RS01765	RDD-like protein domain protein
VY93_RS01760	Transcription antitermination factor NusB
VY93_RS02865	Replication initiation and membrane attachment protein
VY93_RS02705	LMP repeated-region protein
VY93_RS00715	Aromatic cluster surface protein
VY93_RS00710	Aromatic cluster surface protein
VY93_RS00705	Aromatic cluster surface protein
VY93_RS00680	Periplasmic binding protein, Laci family transcriptional regulator, sucrose operon repressor
VY93_RS03280	Chaperonin Cpn60/TCP-1
VY93_RS02250	SMC-like protein; zinc finger protein; SH2 motif-like domain protein
VY93_RS02310	SH2 domain-containing protein
VY93_RS03725	Myosin head, motor region-containing protein
VY93_RS00215	Intracellular septation protein A
VY93_RS02950	Herpesvirus BTRF1 protein
VY93_RS03520	Signal-induced proliferation-associated 1-like protein 2 isoform X4; MARVEL domain-containing protein; cell division factor
VY93_RS03805	RNA-binding S4 domain-containing protein
VY93_RS00640	MtN3 and saliva-related transmembrane protein
VY93_RS03665	Mitochondrial carrier
VY93_RS01690	RPS2 ribosomal protein S2
VY93_RS02790	PotD/PotF family extracellular solute-binding protein; excinuclease ABC; aromatic cluster surface protein
VY93_RS03145	Secretin-receptor family
VY93_RS01565	ThrRS/AlaRS common domain superfamily
VY93_RS01215	Lipocalins
VY93_RS00545	Lipocalins

## Data Availability

The data are available in the article or its Appendix A.

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
