# Peer review of "Hypothetical Proteins of Mycoplasma synoviae Reannotation and Expression Changes Identified via RNA-Sequencing"

_microorganisms, 2023, doi:10.3390/microorganisms11112716_

Round 1

Reviewer 1 Report

Comments and Suggestions for Authors

This manuscript describes an interesting entirely in silico approach for re-annotation of hypothetical proteins using bioinformatics tools and prediction algorithms. While the approach is interesting, I am afraid that without any isolation and actual experimental assessment of the function and/or localization of any of the hypothetical proteins, the annotations cannot be verified and thus, cannot be accepted. In particular, virulence prediction confirmation by RNA seq and differential transcription is insufficient. Also, there are lots of references to unpublished experimental data, yet it is not clear why the authors did not include that data in the manuscript. It is a pity, as this is a well written manuscript, but without any experimental data besides RNA seq, unfortunately, it is my opinion that it cannot be published.

Comments on the Quality of English Language

The language is fine. Minor changes are required. 

Author Response

Computer methods have been widely used in many aspects of the study of microorganisms and their functions. They rely on various algorithms, after a large number of simulation experiments, constantly improve the accuracy rate, and finally have a place in various fields of scientific research. The algorithms of the tools used in this study have been evaluated, and the accuracy of the functional annotation software used is as high as 95%, so the annotation results are trustworthy. These results will contribute to M. synoviae research.

Reviewer 2 Report

Comments and Suggestions for Authors

This manuscript aimed to reannotate the  155 proteins in M. synoviae ATCC 25204 to predict new potential virulence factors using currently available databases and bioinformatics tools

This is a well-written manuscript that provides a clear overview of the study. The authors have clearly stated the purpose of the study, the methods used, the results obtained, and the conclusions drawn.

The abstract is a well-written and provides a clear and concise overview of the study. But, In the first sentence, you could add a brief statement about the importance of Mycoplasma synoviae infection in chickens. In the second paragraph, you could provide more specific information about the methods used to reannotate the proteins. For example, you could mention which databases and bioinformatics tools were used.

The Introduction section is well-written and informative. It provides a good overview of the background information on Mycoplasma synoviae infection, the importance of identifying novel virulence factors, and the research question or hypothesis of the study.

Some specific improvements could be made to the Introduction section. For example, the authors could provide more information about the pathogenesis of M. synoviae infection and the role of virulence factors in this process. Additionally, the authors could state their research question or hypothesis more clearly and be more specific about their methods.

Here are some specific suggestions for improvement:

The authors could say "Mycoplasma synoviae is a devastating pathogen of chickens that costs the poultry industry billions of dollars each year."

In the second paragraph, the authors could provide more information about the pathogenesis of M. synoviae infection and the role of virulence factors in this process. They could also mention the importance of identifying novel virulence factors for developing new diagnostic tools and treatments.

In the third paragraph, the authors state their research question as "The functions of the HPs are unknown." However, they could make this more specific by stating their hypothesis that some of the HPs may be novel virulence factors.

In the fourth paragraph, the authors could mention the specific bioinformatics tools that they used to predict the functions of the HPs and to identify differentially expressed genes. They could also mention how they validated their predictions.

Materials and Methods

Line 75. Provide a hyperlink for the sequence information.

Have the authors analyzed any field strain?

In the section on predicting the virulence of HPs, what is the specific criteria that you used to identify potential virulence factors.

In the section on predicting the antigenicity of HPs, what is the specific algorithm that you used to predict epitopes.

Add more detail about the RNA-Seq library preparation and sequencing methods that you used.

The results and the discussion section are well written.

The figures need to be replaced with another high quality figures.

Author Response

Thank you for your precise comments on our article. According to your suggestions, we have supplemented several data here and corrected several mistakes in our previous draft. Based on your comments, we also attached a point-by-point letter to you. We have made extensive revisions to our previous draft,all changes  were marked in red font. The detailed point-by-point responses are listed below.
The authors could say "Mycoplasma synoviae is a devastating pathogen of chickens that costs the poultry industry billions of dollars each year."
A: This is a good proposal,  the phrase has been added to the newly available manuscript, L30 to L32.
In the second paragraph, the authors could provide more information about the pathogenesis of M. synoviae infection and the role of virulence factors in this process. They could also mention the importance of identifying novel virulence factors for developing new diagnostic tools and treatments.
A: Thank you for your good advice, the relevant literature has been added to the newly submitted manuscript, L59 to L62.
In the third paragraph, the authors state their research question as "The functions of the HPs are unknown." However, they could make this more specific by stating their hypothesis that some of the HPs may be novel virulence factors.
A:  The new virulence factor is relative to the entire genome. It is the virulence factor that already exists throughout the proteome but has not been annotated.
In the fourth paragraph, the authors could mention the specific bioinformatics tools that they used to predict the functions of the HPs and to identify differentially expressed genes. They could also mention how they validated their predictions.
A: After annotating HPs, using the measured transcriptome data, the expression of individual genes of HPs upon exposure to host cells was analyzed. Prokaryotic transcription data has been submitted to NCBI.
Materials and Methods
Line 75. Provide a hyperlink for the sequence information.
A: Hyperlinks are already provided in this section, L81.
Have the authors analyzed any field strain?
A: We thank the referee for this good question. We used the full prokaryotic genome database as reference for the reannotation of putative proteins of M. synoviae ATCC 25204 and did not use any field strains. The main reasons include the following two points: first, the lack of a prokaryotic genome database for M. synoviae; and second, the selection of a relatively large prokaryotic genome to better understand the biological information of putative proteins in M. synoviae.
In the section on predicting the virulence of HPs, what is the specific criteria that you used to identify potential virulence factors.
A: The proteins which evaluated as virulent via VirulentPred software and aligned to known virulence factors with homology by VFDB,These proteins are considered potential virulence factors.
In the section on predicting the antigenicity of HPs, what is the specific algorithm that you used to predict epitopes.
A: Analysis of data from experimentally determined antigenic sites on proteins has revealed that the hydrophobic residues Cys, Leu and Val, if they occur on the surface of a protein, are more likely to be a part of antigenic sites. A semi-empirical method which makes use of physicochemical properties of amino acid residues and their frequencies of occurrence in experimentally known segmental epitopes was developed to predict antigenic determinants on proteins. Application of this method to a large number of proteins has shown that our method can predict antigenic determinants with about 75% accuracy which is better than most of the known methods. This method is based on a single parameter and thus very simple to use.

Reviewer 3 Report

Comments and Suggestions for Authors

The paper with the title “Hypothetical proteins of Mycoplasma synoviae reannotation and expression changes identified via RNA-sequencing” improved the functional annotation of M. synoviae ATCC 25204 21 from 76% to 95% and enabled the discovery of potential virulence factors in the genome. The paper is well written and the results are very convincing.

Suggestions for specific details to be added to the main text are listed below:

L9 and L29: Mycoplasma (M.) synoviae instead of Mycoplasma synoviae

L32: M. synoviae instead of Mycoplasma synoviae

L34-35: M. synoviae in cursive

L39: remove the extra-dot after macrophages

L46: adhegenins, do you mean adhesines?

L63-67: check the numbers: 125 reannotated (includind the 28 virulence protein) + 9 not reanotated are not 155.

L109: in vitro in cursive

L151-153: No mention of the Supplementary Table 5b has been done in the text.

L252: in silico in cursive

L259 and L261: in vitro in cursive

Figures 3, 4 and 5 are far too small

Author Response

Thank you for your precise comments on our article. According to your suggestions, we have supplemented several data here and corrected several mistakes in our previous draft. Based on your comments, we also attached a point-by-point letter to you. We have made extensive revisions to our previous draft,all changes  were marked in red font. The detailed point-by-point responses are listed below.
L9 and L29: Mycoplasma (M.) synoviae instead of Mycoplasma synoviae
A:"Mycoplasma synoviae" at L9 and L29 at the beginning of the sentence,so there is no abbreviation. We could modify it if needed. L9 and L29
L32: M. synoviae instead of Mycoplasma synoviae
"Mycoplasma synoviae" at L32 at the beginning of the sentence,so there is no abbreviation. We could modify it if needed. L33
L34-35: M. synoviae in cursive
A:The font of 'M. synoviae' has been changed to cursive. L35 
L39: remove the extra-dot after macrophages
A:  The extra-dot has been removed.  L44
L46: adhegenins, do you mean adhesines?
A: Sorry, there is a misspelling here. 'adhegenins' has been revised to 'adhesines'.  L47
L63-67: check the numbers: 125 reannotated (includind the 28 virulence protein) + 9 not reanotated are not 155.
A: Sorry for the misrepresentation here. This phrase has been revised to 'Among those HPS, 28 were predicted as virulence proteins, including 19 virulence proteins, 125 were reannotated, and 9 virulence proteins were not reannotated .' L71 to L73
L109:  in cursive
A: The font of  'in vitro' has been changed to cursive. L155
L151-153: No mention of the Supplementary Table 5b has been done in the text.
A: The Supplementary Table 5b statement has been described in the new manuscript. L155 toL159
L252: in cursive
A:The font of ' in silico' has been changed to cursive.L263
L259 and L261: in vitro in cursive
A: The font of 'in vitro' has been changed to cursive. L270 and L272
Figures 3, 4 and 5 are far too small
A: Figures 3, 4 and 5 have been enlarged in new manuscript.

Round 2

Reviewer 1 Report

Comments and Suggestions for Authors

This is a revised manuscript describing the re-annotation of M. synoviae hypothetical proteins using an in silico approach. While I do appreciate the fact that bioinformatics algorithms may have a predicted accuracy of nearly 95%, it is nonetheless a prediction. All of these algorithms are used extensively in research as the authors correctly point out, yet to reannotate proteins experimental validation is still necessary. The authors have indeed attempted to validate some of the predictions by RNA seq, which suggests that they are aware of the fact that predictions do need experimental validation. I am afraid however that RNA seq cannot possibly validate all of the predictions which were made.

Reannotation is important work and bioinformatics tools are a great first step but this cannot stand alone. There must be experimental validation of at least a few of the proteins to confirm function and/or localization. Therefore I must emphasize that this manuscript cannot be published in its current form, as reannotation based on predictions is simply insufficient before it can be acceptable. The authors have done a lot of good work in this manuscript and it is well written, but another very major point is also that the title of the manuscript does not reflect what was actually done.

Comments on the Quality of English Language

English is fine.

Author Response

First, we did not identify virulence factors via RNA-seq, but only observed changes in the expression of hypothetical proteins when exposed to host cells. Since some of the putative proteins were predicted to be virulence factors, the proteins were concerned and analyzed, particularly.
It is important to understand the potential function of the protein before protein function verification. The main work of this paper is to reveal the potential functions of hypothetical proteins, improve the proteome of M. synoviae, and provide data for the study of M. synoviae proteins.